# Urinary, Gastrointestinal, and Sexual Dysfunctions after Chemotherapy, Radiotherapy, Radical Surgery or Multimodal Treatment in Women with Locally Advanced Cervical Cancer: A Multicenter Retrospective Study

**DOI:** 10.3390/cancers15245734

**Published:** 2023-12-07

**Authors:** Mariano Catello Di Donna, Giuseppe Cucinella, Vincenzo Giallombardo, Giulio Sozzi, Nicolò Bizzarri, Giovanni Scambia, Basilio Pecorino, Paolo Scollo, Roberto Berretta, Vito Andrea Capozzi, Antonio Simone Laganà, Vito Chiantera

**Affiliations:** 1Unit of Gynecologic Oncology, ARNAS “Civico-Di Cristina-Benfratelli”, 90127 Palermo, Italy; marianocatello.didonna@unipa.it (M.C.D.D.); vincenzo.giallombardo@asppalermo.org (V.G.); 2Department of Surgical, Oncological and Oral Sciences (Di.Chir.On.S.), University of Palermo, 90133 Palermo, Italy; 3Gynecologic Oncology, Istituto Nazionale Tumori-IRCCS-Fondazione G. Pascale, 80131 Naples, Italy; 4Department of Obstetrics and Gynecology, Fondazione Istituto G. Giglio, 90015 Cefalù, Italy; giuliosozzi@hotmail.it; 5Dipartimento della Salute della Donna, del Bambino e di Sanità Pubblica, Fondazione Policlinico Universitario A. Gemelli IRCCS, 00168 Rome, Italy; 6Gynecologic Oncology Unit, Department of Woman and Child Health and Public Health, Fondazione Policlinico Universitario A. Gemelli IRCCS, Largo Francesco Vito 1, 00168 Rome, Italy; nicolo.bizzarri@policlinicogemelli.it (N.B.); giovanni.scambia@policlinicogemelli.it (G.S.); 7Faculty of Medicine and Surgery, Università Cattolica del Sacro Cuore, Largo Francesco Vito 1, 00168 Rome, Italy; 8Maternal and Child Department, Obstetrics and Gynecology, Cannizzaro Hospital, 95100 Catania, Italy; eliopek@gmail.com (B.P.); paolo.scollo@unikore.it (P.S.); 9Maternal and Child Department, University of Enna “Kore”, 94100 Enna, Italy; 10Department of Obstetrics and Gynecology, University of Parma, 43125 Parma, Italy; rberretta@ao.pr.it (R.B.); vitoandrea.capozzi@studenti.unipr.it (V.A.C.); 11Unit of Obstetrics and Gynecology, “Paolo Giaccone” Hospital, 90127 Palermo, Italy; antoniosimone.lagana@unipa.it; 12Department of Health Promotion, Mother and Child Care, Internal Medicine and Medical Specialties (PROMISE), University of Palermo, 90133 Palermo, Italy

**Keywords:** cervical cancer, quality of life, multimodal treatments, neoadjuvant concurrent chemoradiation, radical surgery, radiotherapy, chemotherapy, patient-reported outcomes

## Abstract

**Simple Summary:**

The quality of life of patients with locally advanced cervical cancer (LACC) is impacted by the treatment they receive. The aim of our retrospective study was to evaluate urinary, bowel, and sexual dysfunction in a series of LACC patients who were treated with chemotherapy, radiotherapy, radical surgery, or a combination of these treatments. In a population of 90 LACC patients, we observed an increase in urinary frequency associated with other urinary dysfunction symptoms in the group receiving exclusive radio–chemotherapy. Overall, 57.8% of patients were not sexually active after treatment, and pain was the main reason for avoiding sexual activity. Although the rate and severity of urinary, gastrointestinal, and sexual dysfunction were similar in the four groups, exclusive radio–chemotherapy was associated with worse sexual and urinary outcomes.

**Abstract:**

Background: Different strategies have been proposed for the treatment of locally advanced cervical cancer (LACC), with different impacts on patient’s quality of life (QoL). This study aimed to analyze urinary, bowel, and sexual dysfunctions in a series of LACC patients who underwent chemotherapy, radiotherapy, radical surgery, or a combination of these treatments. Methods: Patients with LACC who underwent neoadjuvant radio–chemotherapy (NART/CT; *n* = 35), neoadjuvant chemotherapy (NACT; *n* = 17), exclusive radio–chemotherapy (ERT/CT; *n* = 28), or upfront surgery (UPS; *n* = 10) from November 2010 to September 2019 were identified from five oncological referral centers. A customized questionnaire was used for the valuation of urinary, gastrointestinal, and sexual dysfunctions. Results: A total of 90 patients were included. Increased urinary frequency (>8 times/day) was higher in ERT/CT compared with NACT/RT (57.1% vs. 28.6%; *p* = 0.02) and NACT (57.1% vs. 17.6%; *p* = 0.01). The use of sanitary pads for urinary leakage was higher in ERT/CT compared with NACT/RT (42.9% vs. 14.3%; *p* = 0.01) and NACT (42.9% vs. 11.8%; *p* = 0.03). The rate of reduced evacuations (<3 times a week) was less in UPS compared with NACT/RT (50% vs. 97.1%; *p* < 0.01), NACT (50% vs. 88.2, *p* < 0.01), and ERT/CT (50% vs. 96.4%; *p* < 0.01). A total of 52 women were not sexually active after therapy, and pain was the principal reason for the avoidance of sexual activity. Conclusions: The rate and severity of urinary, gastrointestinal, and sexual dysfunction were similar in the four groups of treatment. Nevertheless, ERT/CT was associated with worse sexual and urinary outcomes.

## 1. Introduction

Cervical cancer is the fourth most common malignancy in women and the most frequent cause of death from gynecological cancers worldwide [1]. In Italy, cervical cancer is the fifth most frequently diagnosed cancer in women, with an estimated 2400 new cases in 2020 [2]. Although there is a high chance of preventing the tumor, approximately more than one third of patients present with locally advanced cervical cancer (LACC) at diagnosis.

LACC corresponds to an invasive carcinoma of more than 4 cm in its greatest dimension and/or to a tumor that extends beyond the uterus, including the FIGO (International Federation of Gynecology and Obstetrics) stage from IB3 to IVA [3]. The current standard treatment for patients with LACCs is exclusive primary chemoradiation (ERT/CT), with possible additional vaginal brachytherapy (BRT) [4]. Moreover, an important factor guiding treatment planning is the status of the aortic lymph nodes. Indeed, patients without nodal disease or with disease limited to the pelvis undergo pelvic external beam radiation therapy (EBRT) with concurrent cisplatin-based chemotherapy and BRT. In contrast, for patients with positive para-aortic lymph nodes, stage FIGO IIIC2, extended-field EBRT with concurrent cisplatin-containing chemotherapy, and BRT should be considered. Para-aortic lymph node involvement occurs in approximately 10–50% of LACC patients, and considering the limitations of imaging modalities, para-aortic surgical staging is also an option for these patients. However, this remains a controversial topic among gynecologic oncologists, and the clinical practice varies widely among different centers [4,5,6,7].

Investigational approaches using surgery after CT/RT, with or without concomitant BRT, have been explored with encouraging results [8,9,10,11,12,13]. NACT, followed by a radical hysterectomy, is an interesting alternative option to CT/RT for highly selected LACC patients, especially for those with stage IB3-IIB disease [14,15,16,17,18,19,20].

Advances in the treatment of patients with LACC have led to an ever-increasing number of cervical cancer survivors over time. However, multimodal treatments are associated with side effects that can impair quality of life (QoL). Thus, previous reports have investigated the QoL and emotional distress in cervical cancer survivors who underwent radical surgery either for first diagnosis or recurrent disease [8,21,22,23].

However, despite the recent increasing attention on the topic, there is limited evidence on the impact of multimodal treatment, namely ERT/CT, NACT, NART/CT, upfront radical surgery (UPS) with/without adjuvant therapy, on the autonomic functions and QoL of patients with LACC. For these reasons, this study aims to analyze women’s self-reported urinary, bowel, and sexual dysfunctions in a consecutive series of LACC patients who underwent ERT/CT, NACT, NART/CT, or UPS with/without adjuvant therapy.

## 2. Materials and Methods

### 2.1. Study Design

This is a multicentric retrospective study of prospective-collected data conducted in patients with LACC treated in five institutions: the Department of Women’s and Children’s Health, Agostino Gemelli Foundation University Hospital, Catholic University of the Sacred Heart, Rome, Italy; the Department of Health Sciences and Medicine, University of Molise, Campobasso, Italy; the Unit of Gynecologic Oncology, ARNAS “Civico-Di Cristina-Benfratelli”, University of Palermo, Palermo, Italy; the Department of Gynecology and Obstetrics, University of Parma, Italy; and the Division of Gynecology and Obstetrics, Maternal and Child Department, Cannizzaro Hospital, Catania, Italy.

The design, analysis, interpretation of data, drafting, and revisions were approved by the Institutional Review Board “Comitato Etico Palermo 2” (approval ID: 39; date of approval: 8 April 2021), conform to the Helsinki Declaration, the Committee on Publication Ethics guidelines (http://publicationethics.org/, accessed on 8 April 2021) and the Strengthening the Reporting of Observational Studies in Epidemiology Statement [24] validated by the Enhancing the Quality and Transparency of Health Research Network (www.equator-network.org, accessed on 8 April 2021). The data collected were anonymized, without personal data that could lead to formal identification of the patient. Each patient in this study was informed about the procedures and signed consent to allow data collection and analysis for research purposes. The study was not advertised. No remuneration was offered to the patients to give consent to be enrolled in this study.

All patients with histological diagnosis of cervical cancer FIGO 2018 stage IB3—IVA who received treatment between November 2010 and September 2019 were considered in the analysis. Patients aged between 18 and 90 years were included. All patients included in the current analysis received one of the following treatment strategies: NART/CT, NACT, ERT/CT, or UPS. NART/CT was administered as follows: patients underwent whole pelvic irradiation (EBRT) in 22 fractions (2 Gy/day, totaling 44 Gy) in combination with cisplatin and 5-fluorouracil or cisplatin and taxol. Slightly different schemes of platinum-based chemotherapy (4 vs. 3 cycles) or radiotherapy (total dose from 44 to 56 Gy) have been administered according to the internal Institutional therapeutic protocols. NACT was administered as follows: patients underwent two different schemes—platinum-based, taxol-ifosfamide-cisplatin (TIP), or cisplatin + taxol (CDDP). ERT/CT was administered as follows: patients underwent EBRT (total dose from 44 Gy + boost 46 Gy), concurrent cisplatin-based chemotherapy (CDDP+ Taxol) with BRT. Regarding UPS, patients underwent laparoscopic total mesometrial resection (L-TMMR), laparotomic total mesometrial resection (TMMR), total laparoscopic radical hysterectomy (TLRH), laparotomic radical hysterectomy, vaginal-assisted laparoscopic radical hysterectomy (LARVH), and laparoscopic-assisted radical vaginal hysterectomy (VARLH). Patients with metachronous or synchronous neoplasia, a history of preoperative urinary, intestinal, or sexual dysfunctions, and psychiatric disorders were excluded. Moreover, all women with recurrence were excluded from the final analysis. For all patients, the final stage was clinically assigned after physical examination under anesthesia, preoperative pelvic magnetic resonance imaging (MRI), and, when required, cystoscopy, rectoscopy, and para-aortic lymph node dissection.

Pelvic dysfunctions were classified as urinary, gastrointestinal, and sexual ones. For all patients, pelvic function outcomes were evaluated by recording patients’ self-reported pelvic functions using a customized questionnaire [25]. Complaints were considered to be moderate/severe when referred to by patients as “often” or “daily”. Patients were interviewed at least one month after the end of the treatment(s). Urinary dysfunctions were evaluated as the presence of dysuria, stress incontinence, urge incontinence, sensation of bladder incomplete emptying, recurrent urinary tract infection, urinary frequency (>8 times/day), nycturia (>2 times/night), enuresis, and use the sanitary pads for urinary leakage. We reported the prevalence of several complaints such as constipation, defecation urgency, fecal incontinence, sensation of incomplete bowel emptying, effort during evacuation, and reduced number of evacuations (<3 times a week). Sexual function was evaluated as the presence of sexual activity, vaginal lubrification, vaginal sensation, and pain during intercourse.

### 2.2. Statistical Analysis

Standard summary statistics were used to describe the demographic and clinical characteristics of the study population. Categorical data were compared with the Chi-square test. The Kruskal–Wallis test, followed by Dunn’s post hoc test for pairwise comparison, was used to compare continuous and ordinal variables. All statistical tests were two-sided, and a value of *p* < 0.05 was considered to be statistically significant. Statistical analysis was performed using Statistical Package for Social Science (SPSS) Version 26.

## 3. Results

During the study period, 202 LACC patients received NART/CT, NACT, ERT/CT, or UPS in the five institutions. We attempted to contact all patients with a telephone call, but 112 women (55.4%) refused or did not respond to phone calls or died during the time of the study. Therefore, the final analysis was conducted on a study population of 90 (44.5%) women (Figure 1).

NART/CT was performed in 35 patients (39%), NACT in 17 patients (19%), ERT/CT in 28 women (31%), and UPS in the remaining 10 patients (11%). The clinical-pathologic characteristics of our study population are presented in Table 1. The median age of the overall population was 53 years (29–87), and the median Body Mass Index (BMI) was 24 (range 18–43). No differences among the four groups were observed in terms of age (*p* = 0.17), BMI (*p* = 0.34), FIGO stage (*p* = 0.27), and tumor histotype (*p* = 0.09). High FIGO grade disease (G3) was more prevalent in patients undergoing NACT or UPS, 80% and 82%, respectively, compared to those undergoing NART/CT or ERT/CT, 54.3% and 35.7%, respectively (*p* = 0.01)

### 3.1. Urinary Functional Outcome

The most relevant complaints were urinary frequency (>8 times/day), stress incontinence, nycturia, sanitary pads, a sensation of incomplete bladder emptying, and urge incontinence detected, respectively, in 34 (37.8%), 30 (33.3%), 24 (26.7%), 22 (24.4%), and 16 (17.8%) patients (Table 2).

Patients submitted to ERT/CT were more likely to complain of urinary frequency (*n* = 16, 57.1%; *p* = 0.03) and use of sanitary pads for urinary leakage (*n* = 12, 42.9%; *p* = 0.04) than those who received other treatment modalities.

In particular, ERT/CT was statistically associated with a higher likelihood of worse outcomes, even after the direct comparison with other regimens of systemic treatment.

ERT/CT vs. NART/CT showed significant differences both for urinary frequency and use of sanitary pads (*p* = 0.02 and *p* = 0.01, respectively). ERT/CT vs. NACT showed significant differences both for urinary frequency and sanitary pads (*p* = 0.01 and *p* = 0.03, respectively). A comparison between each different modality in the impact on urinary frequency and sanitary pads is reported in Table 3.

No significant difference was found between all treatment modalities in the other urinary functional outcome; however, we observed a slightly higher prevalence of stress incontinence among NART/CT group (*n* = 14, 40%; *p* = 0.68); a slightly higher rate of urge incontinence in ERT/CT group (*n* = 7, 25%; *p* = 0.32); and a slightly higher prevalence of sensation of bladder incomplete emptying in NACT group (*n* = 7, 41.2%; *p* = 0.07) (Table 2).

### 3.2. Bowel Functional Outcome

The most relevant complaints were reduced number of evacuations (< 3 times a week), sensation of incomplete bowel emptying, constipation, and effort during evacuation detected, respectively, in 81 cases (90%), 26 cases (28.9%), 25 cases (27.8%) and 21 patients (23.3%) (Table 4).

Patients who underwent NACT were more likely to report moderate/severe sensation of incomplete bowel emptying (*n* = 10, 58.8%; *p* = 0.04), while those who had NART/CT were more likely to report alteration in the evacuations (*n* = 34, 97.1%; *p* < 0.01).

In particular, regarding the reduced number of evacuations (< 3 times a week), we observed a significant difference between UPS and NART/CT (50% vs. 97.1%, *p* < 0.01), between UPS and NACT (50% vs. 88.2%; *p* < 0.01) and between UPS and ERT/CT (50% vs. 96.4%; *p* < 0.01). Regarding the sensation of incomplete bowel emptying, we observed a significant difference between NACT and UPS (58.8% vs. 20%; *p* = 0.047), between NACT and ERT/CT (58.8% vs. 17.9%; *p* < 0.01), and between NACT and NART/CT (58.8% vs. 25.7; *p* = 0.02). Comparison between each different modality in the impact on sensation of incomplete bowel emptying and reduced number of evacuations (<3 times a week) is reported in Table 5.

### 3.3. Sexual Functional Outcome

Focusing on sexuality, 52 (57.8%) patients declared to have not been sexually active after therapy, while 38 (42.2%) women were sexually active. Of these, 3 (7.9%) declared having a minimal or painful vaginal sensation during intercourse, 27 patients (71.1%) had vaginal lubrication during intercourse, and 29 (76.3%) referred to having pain during intercourse (Table 6). Of note, a high prevalence of pain during intercourse was noticed in the NACT group (*n* = 8; 100%); however, this was not significantly different compared with the other groups (*p* = 0.24). Regarding the patients who did not have sexual activity, after the exclusion of patients without a partner, the reasons to renounce sexual activity were to be not interested (*n* = 21), followed by other non-specified reasons (*n* = 9) and pain (*n* = 7). In particular, patients who underwent NACT were those more likely to avoid sex for lack of interest (*n* = 6, 66%; *p* = 0.001), while the patients who underwent ERT/CT did not have sexual activity for pain (*n* = 7, 36.8%; *p* = 0.003). No other significant differences in sexual functional outcomes were observed among the four groups (Table 6).

## 4. Discussion

To date, the standard treatment for patients with advanced stages of cervical cancer is external pelvic RT, typically combined with CT plus BRT. A meta-analysis of 13 randomized trials showed that ERT/CT significantly improved 5-year overall disease-free survival (DFS) [hazard ratio (HR) = 0.78, 95% confidence interval (CI) = 0.70–0.87], 5-year lo-co-regional disease-free survival (HR = 0.76, 95%CI = 0.68–0.86), 5-year metastases-free survival (HR = 0.81, 95%CI = 0.72–0.91) and 5-year overall survival (OS) (HR = 0.81, 95%CI = 0.71–0.91) compared to radiotherapy alone [26]. A meta-analysis of six randomized trials, including 1078 patients with early or locally advanced disease, revealed that NACT followed by radical hysterectomy significantly reduced the risk of progression and the risk of death compared to primary radical hysterectomy [13].

However, NACT followed by radical hysterectomy has been demonstrated to be a possible alternative to the standard treatment. Many trials have shown a significant increase in the survival of patients subjected to NACT and radical hysterectomy compared with those treated with RT. The meta-analysis of five randomized trials including patients with LACC showed that NACT plus radical hysterectomy achieved better overall DFS (HR = 0.68, 95%CI = 0.56–0.82), loco-regional DFS (HR = 0.68, 95%CI = 0.56–0.82), metastases-free survival (HR = 0.63; 95%CI = 0.52–0.78) and OS (HR = 0.65, 95%CI = 0.53–0.80) compared to definitive radiotherapy [27]. In particular, the survival rate of the first group of patients is similar to or even better than the second one (70% vs. 60%). In a study by Benedetti Panici et al. [11], NACT arm experienced a significantly better 5-year PFS (59.7% vs. 46.7%, *p* = 0.02) and 5-year OS (64.7% vs. 46.4%, *p* = 0.005) compared to radiotherapy in patients with stage Ib2-IIb disease, but not in those with stage III disease (5-year PFS = 41.9% vs. 36.4%, *p* = 0.29; 5-year OS = 41.6% vs. 36.7%, *p* = 0.36). Based on the 2018 FIGO classification, patients with stage IB1 disease or with stage IB2-IIA1 disease with intact stromal ring should undergo primary radical surgery, and those with IB2-IIA1 disease with disrupted stromal ring or with IB3 disease could undergo either definitive CCRT or NACT followed by radical surgery. These latter treatments should be indicated especially in relatively young women, also for the lower incidence of long-term vaginal toxicity and compromise of sexual life. Patients with stage ≥IIa2 disease should undergo definitive CCRT.

Overall, few reports have so far evaluated the clinical efficacy of NART/CT. Some authors showed more favorable survival outcomes compared with those who only underwent chemoradiation. Moreover, some studies reported high complication rates associated with NACT/RT [16,28]. Therefore, NACT/RT is not widely performed in most countries, and the rate and severity of potential subsequent complications are still controversial.

Considering this uncertain background, some studies have focused on the toxicity and QoL related to the two different therapies as possible criteria for treatment decisions [29,30,31,32]. From this perspective, we acknowledge that this is a crucial point, considering that these patients have a long life expectancy.

Furthermore, although several studies compared the toxicity of chemotherapy, radiotherapy, and radical surgery, few and non-conclusive data exist on the long-term impact on autonomic pelvic function of multimodal treatments in LACC.

Overall, NART/CT, NACT, ERT/CT, and UPS seem to have a relevant impact on pelvic organ function. Regarding urinary dysfunction, in our study, the most relevant complaints were urinary frequency (>8 times/day), stress incontinence, nycturia, use of sanitary pads for urinary leakage, sensation of bladder incomplete emptying, and urge incontinence. Regarding bowel dysfunction, the most relevant complaints were reduced number of evacuations (<3 times a week), sensation of incomplete bowel emptying, constipation, and effort during evacuation. We found a significant difference in urinary frequency (>8 times/day) (*p* = 0.03) and use of sanitary pads (*p* = 0.04). In particular, urinary frequency (>8 times/day) was observed significantly more in ERT/CT compared with NART/CT group (57.1% vs. 28.6%; *p* = 0.02), as well as comparing ERT/CT with NACT (57.1% vs. 17.6%; *p* = 0.01). Regarding the use of sanitary pads for urinary leakage, we observed a significant difference between ERT/CT and NART/CT (42.9% vs. 14.3%; *p* = 0.01) and between ERT/CT and NACT (42.9% vs. 11.8%; *p* = 0.03).

Regarding gastrointestinal dysfunctions, there was a significant difference in the rate of women with a reduced number of evacuations (*p* < 0.01) and sensation of incomplete bowel emptying (*p* = 0.04). In particular, a reduced number of evacuations (<3 times a week) was observed more often in women who underwent UPS compared with NART/CT (50% vs. 97.1%; *p* < 0.01), comparing UPS and NACT (50% vs. 88.2; *p* < 0.01), and finally comparing UPS and ERT/CT (50% vs. 96.4%; *p* < 0.01). Regarding the sensation of incomplete bowel emptying, we found a significant difference between NACT and UPS (58.8% vs. 20%; *p* = 0.047), between NACT and ERT/CT (58.8% vs. 17.9%; *p* < 0.01), and between NACT and NART/CT (58.8% vs. 25.7; *p* = 0.02). Regarding sexual functional outcome, there was a higher prevalence (53.8%; *p* = 0.15) of reduced vaginal lubrification in the NART/CT group, a higher prevalence of pain during intercourse (100%; *p* = 0.24) in the NACT group, and, finally, a slightly higher prevalence of minimal or painful vaginal sensation in UPS group (25%; *p* = 0.98). Nevertheless, considering sexuality, we acknowledge that 52 patients declared to have not been sexually active after therapy: in particular, among women who underwent ERT/CT, pain was the principal reason for not having sexual activity; in the other three groups, sexual activity was not strictly related to the treatment, but probably to psychological and/or other reasons related to the partner. A final consideration regarding ERT/CT and toxicity is that brachytherapy is an integral part of this treatment. Indeed, uterine position and axis are essential to depth-dose parameters in high-dose-rate brachytherapy, affecting radiation doses received by the bladder and rectum [33]. Furthermore, a dosimetric evaluation study has shown that volumetric treatment planning correlates with some of the side effects of treatment, and 3D dose optimization leads to a form of brachytherapy that is adapted to the individual clinical situation [34].

### Strength and Limitations

Although the homogeneity of the treatments and clinicopathological characteristics at diagnosis in the four investigated groups support the reliability of our findings, several limitations should be taken into account for a proper data interpretation: among the most important ones, we acknowledge that the number of enrolled women is relatively low; as a corollary limitation of the previous one, it was not possible to match the four cohorts for individual patient variables, due to the different number of women included in each group; another important potential bias is that some different surgical approaches were used in women who underwent surgery; in addition, we fairly highlight that we did not use standardized scores/scales, so our data analysis may be considered with low generalizability and comparability with other similar studies; finally, the retrospective nature of the investigation could be considered an intrinsic limitation.

## 5. Conclusions

To the best of our knowledge, this is one of the few studies evaluating autonomic function after four different strategies of treatment for LACC: from this perspective, this data analysis may solicit further investigation on the topic, aimed to provide more robust, evidence-based and tailored support for each subtype of treatment (or combination of treatments). We remark that our study was not aimed to identify the best treatment in women with LACC but rather to give further insights into the effects of different treatments on autonomic functions (namely bladder, bowel, and sexual function) and quality of life. Overall, women undergoing ERT/CT have a high prevalence of urge incontinence, urinary frequency, and use of sanitary pads for urinary leakage and pain, which was identified as the principal reason for refusing sexual activity; conversely, women undergoing NACT have a high prevalence of sensation of incomplete bowel emptying. In synthesis, ERT/CT has been associated with worse sexual and urinary outcomes, while NACT has worse bowel functional outcomes.

## Figures and Tables

**Figure 1 cancers-15-05734-f001:**
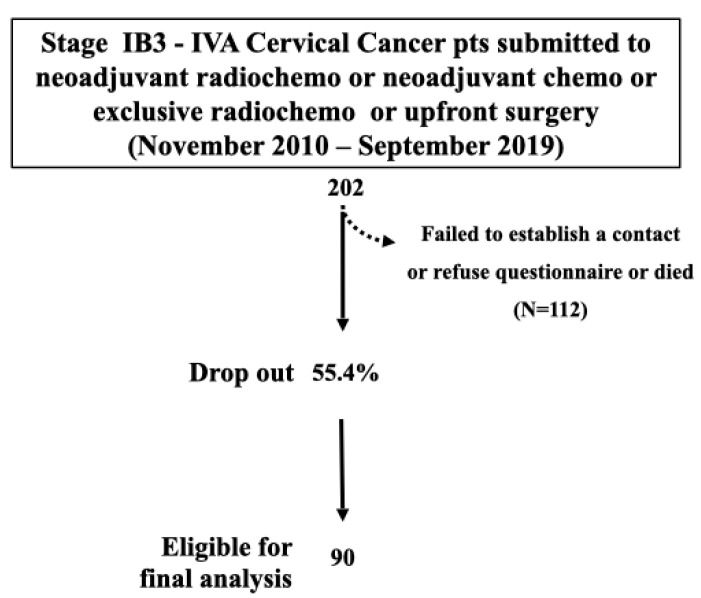
Study flowchart.

**Table 1 cancers-15-05734-t001:** Clinicopathological characteristics of the study population.

Characteristics	All Cases	NART/CT	NACT	ERT/CT	UPS	*p*-Value *
All cases	90	35	17	28	10	
Age, median (range) †	53 (29–87)	54 (29–84)	52 (34–84)	55.5 (29–87)	47 (39–66)	0.17
BMI, median (range) †	24 (18–43)	26.5 (18–35)	22 (18–41)	23 (20–43)	25.5 (19–33)	0.34
Tumor Grade						
G1	1 (1.1)	0 (0)	0 (0)	1 (3.6)	0 (0)	0.01
G2	38 (42.2)	16 (45.7)	3 (17.6)	17 (60.7)	2 (20)	
G3	51 (56.7)	19 (54.3)	14 (82.4)	10 (35.7)	8 (80)	
FIGO Stage						
IB 3/IIA 2	22 (24.4)	6 (17.1)	4 (23.5)	5 (17.9)	7 (70)	0.27
II B	43 (47.8)	21 (60)	8 (47.1)	14 (50)	0 (0)	
III A/III B	8 (8.9)	4 (11.4)	1 (5.9)	3 (10.7)	0 (0)	
IIIC	10 (11.1)	1 (2.9)	2 (11.8)	4 (14.3)	3 (30)	
IV A	7 (7.8)	3 (8.6)	2 (11.8)	2 (7.1)	0 (0)	
Tumor Histotype						
Squamous cell	74 (82.2)	29 (82.9)	11 (64.7)	25 (89.3)	6 (60)	0.09
Adenocarcinoma	10 (11.1)	5 (14.3)	2 (11.8)	0 (0)	3 (30)	
Others	6 (6.7)	1 (2.9)	4 (23.5)	3 (10.7)	1 (10)	

NART/CT: Neoadjuvant radio–chemotherapy, NACT: neoadjuvant chemotherapy, ERT/CT: Exclusive chemotherapy, UPS: upfront surgery. FIGO: International Federation of Gynecology and Obstetrics. * Calculated using χ2 exact test; † Calculated using Kruskal–Wallis test.

**Table 2 cancers-15-05734-t002:** Urinary functional outcome of the study population.

Urinary Functional Outcome	All Cases Nr (%)	NART/CT Nr (%)	NACT Nr (%)	ERT/CT Nr (%)	UPS Nr (%)	*p*-Value *
Dysuria (moderate/severe)	4 (4.4)	2 (5.7)	1 (5.9)	1 (3.6)	0 (0)	0.86
Stress incontinence (moderate/severe)	30 (33.3)	14 (40)	5 (29.4)	8 (28.6)	3 (30)	0.68
Urge incontinence (moderate/severe)	16 (17.8)	7 (20)	2 (11.8)	7 (25)	0 (0)	0.32
Sensation of incomplete bladder emptying (often/daily)	21 (23.3)	7 (20)	7 (41.2)	7 (25)	0 (0)	0.07
Recurrent urinary tract infection (>4 times/year)	4 (4.4)	0 (0)	1 (5.9)	3 (10.7)	0 (0)	0.19
Urinary frequency (>8 times/day)	34 (37.8)	10 (28.6)	3 (17.6)	16 (57.1)	5 (50)	0.03
Nycturia (moderate/severe)	24 (26.7)	10 (28.6)	5 (29.4)	7 (25)	2 (20)	0.92
Enuresis (often/daily)	7 (7.8)	4 (11.4)	1 (5.9)	2 (7.1)	0 (0)	0.67
Sanitary pads (often/daily)	22 (24.4)	5 (14.3)	2 (11.8)	12 (42.9)	3 (30)	0.04

NART/CT: Neoadjuvant radio–chemotherapy, NACT: neoadjuvant chemotherapy, ERT/CT: Exclusive chemotherapy. UPS: upfront surgery. * Kruskal–Wallis test used.

**Table 3 cancers-15-05734-t003:** Comparison between each different modality in the evaluation of urinary frequency (>8 times/day) and use of sanitary pads (Often/Daily).

Urinary Frequency (>8 Times/Day)	*p*-Value *
NART/CT vs. NACT	0.5
NART/CT vs. ERT/CT	0.02
NART/CT vs. UPS	0.22
NACT vs. ERT/CT	0.01
NACT vs. UPS	0.11
ERT/CT vs. UPS	0.39
Sanitary Pads (Often/Daily)	
NART/CT vs. NACT	0.89
NART/CT vs. ERT/CT	0.01
NART/CT vs. UPS	0.25
NACT vs. ERT/CT	0.03
NACT vs. UPS	0.31
ERT/CT vs. UPS	0.42

NART/CT: Neoadjuvant radio–chemotherapy, NACT: neoadjuvant chemotherapy. ERT/CT: Exclusive chemotherapy, UPS: upfront surgery. * Dunn’s post hoc test was used.

**Table 4 cancers-15-05734-t004:** Bowel functional outcome of the study population.

Gastrointestinal Dysfunctions	All Cases Nr (%)	NART/CT Nr (%)	NACT Nr (%)	ERT/CT Nr (%)	UPS Nr (%)	*p*-Value *
Constipation (moderate/severe)	25 (27.8)	9 (25.7)	7 (41.2)	8 (28.6)	1 (10)	0.3
Defecation urgency (moderate/severe)	14 (15.6)	5 (14.3)	2 (11.8)	6 (21.4)	1 (10)	0.78
Fecal incontinence (moderate/severe)	2 (2.2)	2 (5.7)	0 (0)	0 (0)	0 (0)	0.37
Sensation of incomplete bowel emptying (moderate/severe)	26 (28.9)	9 (25.7)	10 (58.8)	5 (17.9)	2 (20)	0.04
Effort during evacuation	21 (23.3)	8 (22.9)	5 (29.4)	5 (17.9)	3 (30)	0.74
Evacuations (<3 times a week)	81 (90)	34 (97.1)	15 (88.2)	27 (96.4)	5 (50)	<0.01

NART/CT: Neoadjuvant radio–chemotherapy, NACT: neoadjuvant chemotherapy, ERT/CT: Exclusive chemotherapy. UPS: upfront surgery. * Kruskal–Wallis test used.

**Table 5 cancers-15-05734-t005:** Comparison between each different modality in the evaluation of sensation of incomplete bowel emptying and reduced number of evacuations (<3 times a week).

Sensation of Incomplete Bowel Emptying	*p*-Value *
Neoadjuvant CT/RT vs. NACT	0.02
Neoadjuvant CT/RT vs. Exclusive CT/RT	0.89
Neoadjuvant CT/RT vs. Upfront Surgery	0.72
NACT vs. Exclusive CT/RT	<0.01
NACT vs. Upfront Surgery	0.047
Exclusive CT/RT vs. Upfront Surgery	0.89
Evacuations (<3 times a week)	
Neoadjuvant CT/RT vs. NACT	0.29
Neoadjuvant CT/RT vs. Exclusive CT/RT	0.92
Neoadjuvant CT/RT vs. Upfront Surgery	<0.01
NACT vs. Exclusive CT/RT	0.35
NACT vs. Upfront Surgery	<0.01
Exclusive CT/RT vs. Upfront Surgery	<0.01

NART/CT: Neoadjuvant radio–chemotherapy, NACT: neoadjuvant chemotherapy, ERT/CT: Exclusive chemotherapy, UPS: upfront surgery. * Dunn’s post hoc test was used.

**Table 6 cancers-15-05734-t006:** Sexual functional outcome of the study population.

Characteristics	All Cases Nr (%)	NART/CT Nr (%)	NACT Nr (%)	ERT/CT Nr (%)	UPS Nr (%)	*p*-Value *
Sexual Activity						
Yes	38 (42.2)	13 (37.1)	8 (47.1)	9 (32.1)	8 (80)	0.06
No	52 (57.8)	22 (62.9)	9 (52.9)	19 (67.9)	2 (20)	
Reasons for No Sexual Activity						
Not interested	21 (40.4)	13 (59.1)	6 (66.7)	2 (10.5)	0 (0)	0.001
Pain	7 (13.5)	0 (0)	0 (0)	7 (36.8)	0 (0)	0.003
Other	9 (17.3)	5 (22.7)	2 (22.2)	1 (5.3)	1 (50)	0.168
Vaginal Lubrification						
Yes	27 (71.1)	6 (46.2)	7 (87.5)	8 (88.9)	6 (75)	0.15
No	11 (28.9)	7 (53.8)	1 (12.5)	1 (11.1)	2 (25)	
Vaginal Sensation						
Normal	35 (92.1)	13 (100)	8 (100)	8 (88.9)	6 (75)	0.98
Minimal/painful	3 (7.9)	0 (0)	0 (0)	1 (11.1)	2 (25)	
Pain During Intercourse						
Yes	29 (76.3)	8 (61.5)	8 (100)	6 (66.7)	7 (87.5)	0.24
No	9 (23.7)	5 (38.5)	0 (0)	3 (33.3)	1 (12.5)	

NART/CT: Neoadjuvant radio–chemotherapy, NACT: neoadjuvant chemotherapy. ERT/CT: Exclusive chemotherapy, UPS: upfront surgery. * Calculated using χ2 exact test and Kruskal–Wallis test, as appropriate.

## Data Availability

The data used and materials to support the findings of this study are available from the corresponding author upon request.

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
