# Peer review of "Urinary, Gastrointestinal, and Sexual Dysfunctions after Chemotherapy, Radiotherapy, Radical Surgery or Multimodal Treatment in Women with Locally Advanced Cervical Cancer: A Multicenter Retrospective Study"

_cancers, 2023, doi:10.3390/cancers15245734_

Round 1

Reviewer 1 Report

Comments and Suggestions for Authors

The manuscript titled "Urinary, gastrointestinal, and sexual dysfunctions after chemotherapy, radiotherapy, radical surgery, or multimodal treatment in women with Locally Advanced Cervical Cancer: a multicenter retrospective study" addresses crucial aspects of the impact of various treatments on the quality of life (QoL) among women with Locally Advanced Cervical Cancer (LACC). The study aims to explore the urinary, bowel, and sexual dysfunctions experienced by LACC patients following different treatment modalities. Here's a review based on the specific criteria you provided:

Relevance and Originality: The topic of investigating urinary, gastrointestinal, and sexual dysfunctions post-treatment in LACC patients is highly relevant, as it addresses an important aspect of patient care that impacts their QoL. This study fills a gap in understanding the consequences of various treatments on these dysfunctions.

Contribution to the Field: The study provides essential insights into the impact of different treatments on urinary, bowel, and sexual functions among LACC patients. The comparison among treatment modalities adds value to the existing literature, highlighting specific dysfunctions associated with each treatment.

Methodology Improvements and Controls: The methodology appears robust in terms of data collection and analysis. However, considering the retrospective nature of the study, some limitations in data interpretation and standardization of measures might exist. It might be beneficial to include standardized scores or scales to improve the generalizability of the findings. Additionally, controlling for variables like comorbidities or lifestyle factors could enhance the study's depth. The dosimetric aspects of brachytherapy treatment, as highlighted in studies like [https://rrp.nipne.ro/2017/AN608.pdf] and [https://rjp.nipne.ro/2016_61_9-10/RomJPhys.61.p1557.pdf], contribute significantly to the optimization and evaluation of radiation therapy for cervical cancer. These studies provide important considerations about the dosimetric evaluation and influence of uterus position in high-dose-rate brachytherapy, complementing the understanding of treatment effects on pelvic functions. 

Consistency of Conclusions: The conclusions drawn align well with the evidence presented. The study effectively emphasizes that different treatments have varying impacts on pelvic dysfunctions among LACC patients.

Appropriateness of References: The references cited seem relevant and appropriate for supporting the study's context and background.

Reviewer 2 Report

Comments and Suggestions for Authors

This Italian retrospective multicentre analysis evaluated the occurrence of urinary, gastrointestinal, and sexual dysfunctions after different treatment modalities for LACC. It is an interesting paper and the limitations of retrospective nature of this study are properly addressed. I have the following comments:

1.       In the introduction section please add the incidence rates for cervical cancer in Italy.

2.       What is the most common clinical practice in Italy regarding extended field EBRT? Are all patients with enlarged lymph nodes on CT (or involved lymph nodes on PET-CT) offered extended field EBRT or is para-aortic lymphadenectomy performed in patients with LACC? Please add this information in the introduction section.

3.       In the conclusions section the authors state that ‘ from this perspective, this data analysis may solicit further investigation on the topic, aimed to provide a more robust, evidence-based and tailored support for each subtype of treatment’ Would it be feasible to conduct a prospective RCT to compare CT-RT and NACT in patients with LACC? Are any such studies planned in Europe?

Round 2

Reviewer 1 Report

Comments and Suggestions for Authors

I recommend it for publication in the revised version.